# Exosome/Liposome-like Nanoparticles: New Carriers for CRISPR Genome Editing in Plants

**DOI:** 10.3390/ijms22147456

**Published:** 2021-07-12

**Authors:** Mousa A. Alghuthaymi, Aftab Ahmad, Zulqurnain Khan, Sultan Habibullah Khan, Farah K. Ahmed, Sajid Faiz, Eugenie Nepovimova, Kamil Kuča, Kamel A. Abd-Elsalam

**Affiliations:** 1Biology Department, Science and Humanities College, Shaqra University, Alquwayiyah 19245, Saudi Arabia; malghuthaymi@su.edu.sa; 2Department of Biochemistry, University of Agriculture, Faisalabad 38040, Pakistan; aftab.ahmad@uaf.edu.pk; 3Center for Advanced Studies in Agriculture and Food Security (CASAFS), University of Agriculture, Faisalabad 38040, Pakistan; sultan@uaf.edu.pk; 4Institute of Plant Breeding and Biotechnology, MNS University of Agriculture, Multan 60000, Pakistan; zulqurnain.khan@mnsuam.edu.pk; 5Biotechnology English Program, Faculty of Agriculture, Cairo University, Giza 12613, Egypt; farahkamel777@gmail.com; 6Department of Plant Breeding and Genetics, University of Haripur, Haripur 22620, Pakistan; sfiaz@uoh.edu.pk; 7Department of Chemistry, Faculty of Science, University of Hradec Kralove, 50003 Hradec Kralove, Czech Republic; eugenie.nepovimova@uhk.cz; 8Agricultural Research Center (ARC), Plant Pathology Research Institute, 9-Gamaa St., Giza 12619, Egypt

**Keywords:** genome editing, CRISPR, nanoparticles, exosomes and liposomes

## Abstract

Rapid developments in the field of plant genome editing using clustered regularly interspaced short palindromic repeats (CRISPR)/CRISPR-associated protein (Cas) systems necessitate more detailed consideration of the delivery of the CRISPR system into plants. Successful and safe editing of plant genomes is partly based on efficient delivery of the CRISPR system. Along with the use of plasmids and viral vectors as cargo material for genome editing, non-viral vectors have also been considered for delivery purposes. These non-viral vectors can be made of a variety of materials, including inorganic nanoparticles, carbon nanotubes, liposomes, and protein- and peptide-based nanoparticles, as well as nanoscale polymeric materials. They have a decreased immune response, an advantage over viral vectors, and offer additional flexibility in their design, allowing them to be functionalized and targeted to specific sites in a biological system with low cytotoxicity. This review is dedicated to describing the delivery methods of CRISPR system into plants with emphasis on the use of non-viral vectors.

## 1. Introduction

Genetic engineering of plants is at the core of environmental sustainability efforts, natural product synthesis for pharmaceuticals, and agricultural crop engineering and can help meet the needs of the growing human population under changing global climatic conditions [1,2]. Genetic transformation allows the transfer of a foreign gene of interest (a transgene), encoding a trait into the plant cell and introduce the desired trait into a crop. For more success in the genetic engineering of plants, new methods are needed to allow passive transport of diverse biomolecules into a larger range of plant species. Gene vectors, constructed to transfer a gene of interest into a host cell, play an important role in genetic transformations [3].

Nanotechnology-based biotransformation is relatively efficient; however, its combination with commonly applied approaches in transgenic plant development could improve transformation efficiency, productivity, and the chances of minimizing transgene silencing. Nanoparticle-mediated gene transfer depends not only on the defined nanoparticle size and shape but also on the surface functionalization of nanoparticles, nucleic acid protection ability, and biocompatibility [4].

Different forms of magnetic nanoparticles and carbon nanotubes have been used for the development of a sustainable plant transformation method. Various nanostructures hold impressive possibilities for the improvement of our expertise and methodology and the enhancement of plant genetic transformation, specifically in molecular plant-breeding programs [5,6,7]. Modification of physicochemical properties of nanocarriers, protoplast, and plasma membrane could significantly improve liposomal- and nanoparticle-encapsulated gene delivery into the plant cell and help to establish gene transfer methods as a sole process rather than adjuvant [3].

For transfection, a number of nanoparticle-based delivery cargos are available, as opposed to the few methods of gene transformation in plants. For the transformation of plastids or mitochondria especially, which is not possible through agrobacterium-mediated transformation, other vector delivery methods are required. Nanoparticle-based delivery could be successful in bridging the species limitation in plants. Moreover, in the future, new delivery tools could also help carry and transform molecules with a higher molecular weight i.e., proteins. Furthermore, production of non-GMOs to address regulatory concerns would also be a tremendous future achievement. Nanomaterial-based cell membrane vectors can deliver drugs while keeping their configuration and folding in their native states and maintaining stability [8].

Exosomes are naturally occurring signal-transducing membrane-bounded structures that can easily and very precisely deliver the cargos of desired molecular packages in an intercellular framework. Such systems can be useful in forming a nanoplatform for targeted vector delivery and its monitoring [9]. Engineered exosomes with tagged molecules targeted for delivery are generated from synthetic plasma membranes that have extruding mechanisms, very similar to those generated by the real cells [10]. These synthetic exosomes are very important in transporting large proteins and therapeutic genes, working as both endogenous and exogenous vectors [11,12]. Similarly, virus mimetic vesicles have also been developed for the natural virus budding mechanism that occurs upon introduction into an affected cell, resulting in the combination of the host cell membrane and viral surface glycoproteins to generate a lipid envelope that provides an anchorage onto new cells [13]. For transient expression of heterologous proteins in plants, plant viral vectors have been used, such as the tobacco mosaic virus, potato virus X, and cowpea mosaic virus [14]. However, viral vectors are only compatible with selected plant species and expression cassette sizes, which limits the potential plant hosts and hinders expression of large or multiple proteins simultaneously. Additionally, the use of viral vectors, even if used for transient expression of gene editing systems, is usually subject to regulatory purview because of the pathogenic origin of viruses and the fact that viruses may integrate portions of their genetic material into the plant host genome [15]. In general, exosomes have a wide range of potential applications, not only in therapeutics and the designated delivery of target cargos but also in specific applications where cell-to-cell interaction and cell identification is important [16]. This review describes the CRISPR system delivery methods in plants with a focus on the use of non-viral vectors.

## 2. Reagents in CRISPR/Cas

CRISPR/Cas reagents can be inserted into cells as mRNA, DNA, or protein and any of these formats can be employed for electroporation, transfection, and microinjection. In this section, the three reagents of CRISPR/Cas, i.e., RNA, DNA and protein, are discussed. 

### 2.1. DNA

In the case of DNA delivery, Cas9 is carried on a plasmid and expressed under the promoter control. When Cas9 DNA is co-expressed with sgRNA, gene editing can proceed at efficiencies of more than 70%, as calculated by the intensity of indels [17]. The utility of DNA for Cas9 and sgRNA deliveries is that it has the convenience of providing a stable, reusable resource for these reagents, since plasmid DNA can be multiplied in bacteria practically without limitation. Indeed, ordering of plasmids for the DNA-based delivery of sgRNA and Cas9 is a preferred method. Scientists can easily search for a particular gene on specific websites and select from a range of available vectors. In particular, the use of the lentivirus as a means of delivering DNA to transfection-resistant cells is an effective method for producing stable Cas9 and sgRNA cell lines [18].

### 2.2. Protein

Despite the advantages of using DNA for the transmission of CRISPR reagents, numerous research papers have indicated that using Cas9 protein in an RNP complex with sgRNA may provide better genome editing performance. Researchers looked at both the on-target and off-target editing performance of the Cas9 protein and matched it with that of plasmid DNA delivery [17]. Using sgRNA targeting the *CCR5* gene formed by in vitro transcription, scientists electroporated RNP into K562 cells and reported an indel formation rate of 41 percent compared to 50 percent from DNA delivery. Moreover, it is difficult to transfect cells such as BJ fibroblasts and H9 embryonic stem (ES) cells. These researchers observed that RNP on-target editing efficiency was higher than with plasmid DNA delivery, with an estimated indel frequency of 21 percent triggered by Cas9-sgRNA ribonucleoprotein particles (RNPs) compared to an average of 10 percent for plasmids. It was also observed that ES cells, when transfected with Cas9-sgRNA RNPs, developed about twice as many colonies as those transfected with plasmid DNA, indicating that RNP transfection is less harmful to these cells than DNA transfection. However, the explanations for this obvious disparity in toxicity remain unclear.

Besides its on-target effectiveness, SpCas9 can also cut at off-target positions due to its ability to accept non-Watson–Crick base pairing between single-guide RNA (sgRNA) and the target DNA [19]. Off-target cleavage can lead to unintended mutations in these genomic regions. Scientists compared the off-target events of plasmid- and protein-based delivery of Cas9 and observed that, for three different sgRNAs, the percentages of indel mutations at four predicted off-target sites were two to ten times lower when Cas9 was delivered through RNP compared to through plasmids [17]. The lower off-targeting level found for Cas9 RNPs can be explained by its reduced survival in cells, as it was observed that Cas9 delivered via RNP modifies sites much quicker and is deteriorated more quickly than Cas9 delivered via a plasmid.

The Cas9 protein is an enzyme that cuts off foreign DNA as required. Usually, the protein binds to the crRNA (crispr RNA) and tracrRNA (trans-activating crispr RNA). These then lead Cas9 to the target location for slicing. This extension of DNA is complementary to a twenty-nucleotide segment of the crRNA. Using two different regions or domains on its surface, Cas9 breaks both strands of the DNA double helix, which is referred to as a double-stranded break (DSB) [20].

There is an advanced protection system to guarantee that Cas9 does not just cut at undesired locations in the genome. Short sequences of DNA known as protospacer adjacent motifs (PAMs) act as tags and reside adjacent to the target DNA sequence. The Cas9 complex will not cut with PAMs next to a targeted DNA sequence. This is one potential explanation for why Cas9 never targets the CRISPR region of bacteria. By modifying the nucleotide sequence of the RNA guide, an artificial Cas9 framework could be designed to target any DNA sequence for cutting. Cas protein has been altered to make programmable transcription factors that enabled researchers to target and trigger or silence particular genes [21].

Cas12a revealed a variety of primary differences from Cas9, including the staggering cut, depending on the presence of thymine-rich PAMs, in double-stranded DNA it caused compared to the blunt cut created by Cas9, requiring only crRNA for effective targeting [22]. In comparison, Cas9 needs both crRNA and transactive crRNA (tracrRNA).

These differences give Cas12a some benefit over Cas9, i.e., Cas12a small crRNA are suitable for multiplexed genome editing, since more of them can be bundled in one vector than Cas9 sgRNA [23]. In addition, the sticky 5′ overhang left by Cas12a can also be utilized for DNA assembly that is far more target-specific than conventional restriction enzyme cloning. Ultimately, Cas12a cleaves 18–23 base pairs of DNA downstream from the PAM site.

This ensures that there is no damage to the recognition chain after repair; thus, Cas12a facilitates several rounds of DNA cleavage. By comparison, Cas9 cuts just three base pairs upstream of the PAM site, the non-homologous end-joining pathway resulting in the formation of indel mutations that break the recognition chain, preventing further cutting. In principle, repetitive DNA cleavage rounds give rise to increased potential for the desired genomic editing. The distinctive characteristic of Cas12a, relative to Cas9, is that after chopping its target, Cas12a stays attached to the target and only non-discriminately cleaves other ssDNA molecules. This characteristic is known as collateral cleavage and it is used for the production of various diagnostic technologies.

Cas13 is an RNA-guided endonuclease, which implies that it does not cut DNA but rather single-stranded RNA [24]. Cas13 is directed to the ssRNA target by its crRNA and binds and cleaves the target. Similar to Cas12a, Cas13 stays attached to the target and later non-discriminately cleaves other ssRNA molecules. This collateral cleavage characteristic has been used for the production of various diagnostic technologies.

### 2.3. mRNA

In most CRISPR-mediated genome editing technologies, two short RNAs (crRNA and tracrRNA) have been simplified into a single RNA sequence known as the gRNA or into a single-guide RNA (sgRNA). The crRNA portion of the gRNA is a changeable component that contributes to the precision of each CRISPR experiment. As the name suggests, sgRNA is a single RNA molecule that includes the custom-designed short-sequenced crRNA merged into the tracrRNA sequence of the scaffold. The sgRNA may be synthesized or generated in vivo or in vitro from a desired template DNA sequence.

However, some researchers also use guide RNAs with different crRNA and tracrRNA components, which are usually referred to as two-piece gRNAs or simply as cr:tracrRNAs (pronounced as CRISPR tracer RNAs). Synthetic cr:tracrRNA kits are also available for this purpose (https://www.synthego.com/products/crispr-kits/crrna-tracrrna accessed on 3 February 2021). A number of software tools are available to design an optimal guide RNA sequence with minimum off-target effects and maximum on-target efficiency. They include tools such as Desktop Genetics (https://www.crunchbase.com/organization/desktop-genetics accessed on 3 February 2021), Synthego Design Tool (https://www.synthego.com/products/bioinformatics/crispr-design-tool accessed on 3 February 2021), and Benchling (https://www.benchling.com/ accessed on 3 February 2021).

## 3. Carrier Methods in Plants

There are currently several approaches for delivering CRISPR both in vivo and in vitro. Direct and indirect approaches are commonly used for the effective delivery of CRISPR/Cas9-mediated genome editing. Delivery vectors, such as viral and non-viral vectors, can support mRNA or plasmids by instigating nuclease expression in target cells. Alternatively, physical approaches, including lasers, electroporation, ballistic delivery, and microinjection, can also be used to transfer nuclease to cells. Essentially, non-viral vectors are advantageous because there are drawbacks of viral vectors, such as carcinogenesis, reduced encapsulation capacity, and immunogenicity. The delivery methods of CRISPR/Cas reagents are summarized in Figure 1. Moreover, Figure 2 explains the delivery cargos of CRISPR/Cas in plants. 

### 3.1. Direct Methods

Direct gene delivery methods are mostly used for the transient expression of the genes. Stable transformation may also be undertaken using direct methods. A number of researchers prefer the direct gene transfer method owing to the higher transformation efficiencies compared to indirect methods. The direct carrier methods in plants are described below.

#### 3.1.1. PEG-Mediated Delivery

This kind of delivery is undertaken in the presence of polyethylene glycol (PEG). This approach has been successfully extended to protoplasts of various plants, such as *Zea mays*, *Glycine max*, and *Nicotiana tabacum* [25]. The plasmid-containing gRNA and Cas9 are incubated with protoplast in the presence of PEG. PEG-mediated CRISPR was first reported in maize utilizing U3 and CaMV35S promoters for gRNA and Cas9, respectively [26]. In a few experiments, Cas9 was expressed under specific promoters engineered for specific plants and expected to target essential genes. Plasmid DNA was dissolved in water, purified to make it clean, and combined with a suspension of protoplast. After some time, the necessary concentration of PEG was slowly mixed with protoplasts. After that, the protoplasts were regenerated with an effective regeneration medium. Cas9/gRNA ribonucleoproteins were used for the development of transgene-free mutant plants in *Zea may*, *Nicotiana tabacum*, and *Lactuca sativa* using PEG-mediated delivery. The editing frequency for *Lactuca sativa* mutants was up to 46 percent [27].

This key feature of the PEG-mediated delivery system is commonly used to deliver Cas9/gRNA ribonucleoproteins. Vector-less transformed plant mutants are more permissible and potentially not subject to regulatory or ethical barriers. Transgene-free editing of six polyphenol oxidase mushrooms using ribonucleoproteins and PEG was undertaken in 2017. The mutant mushroom showed a 30 percent decrease in the enzyme activity liable for browning and the process also helped it avoid US regulations for genome-edited crops [28]. The ribonucleoprotein complexes cannot be generated by the commonly used Agrobacterium-mediated or floral-dip processes.

#### 3.1.2. Bombardment-Mediated Delivery

This method of transformation involves a gene gun. Generally, silver, tungsten, or gold particles are used as vector carriers. CRISPR/Cas9 components are passed by coated particles to explants by applying high pressure. This approach includes optimum parameters such as target distance, helium pressure, type of explant used, and particle size. Transformed explants are regenerated into a replication medium with a sufficient selection strain.

Successful delivery of Cas9/gRNA ribonucleoproteins and subsequent regeneration of mutants in *Solanum tuberosum*, *Zea may*, and *brassica* have been reported [29,30]. Due to vector less editing, the delivery of Cas9/gRNA ribonucleoprotein by this method is popular. Selection pressure and regeneration of transformed tissues are time-consuming and, thus, very poor editing performance was achieved in maize at just 2.4 to 9.7 per cent [29]. 

The key strength of this method is that there is no requirement for a CRISPR/Cas9 binary vector. Several forms of explants may be transformed with varied and numerous DNA. The most significant of these is the cas9/gRNA ribonucleoprotein complex, which can also be delivered effectively. The key downsides of this method are the random incorporation patterns within the genome, comparatively less editing effectiveness, a higher cost than other methods, and the fact that sites such as the nucleus, cytoplasm, plastid, and mitochondria cannot be controlled.

### 3.2. Indirect Methods

A number of indirect carrier methods are used in plants. These methods are based on *Agrobacterium-mediated* transformation. These methods may be used for transient as well as stable transformation in plants. 

#### 3.2.1. Floral Dip Method

Previously, plasmids were either directly added to the stigma surface or combined with pollen and then introduced to their receptive stigma. In addition, diverse parameters have been designed for effective gene transfer, e.g., dipping and wounding of flowers with female and male organs into suspensions of *Agrobacterium*. The stage of plants is essential for the successful floral transition of plants.

In addition to various physical parameters, such as pH, media composition, optical density, and temperature, molecular factors involving gene size, promoter preference, and vector types are also significant. The well-known constitutive promoters, such as CaMV35S and *Arabidopsis* UBI10, have also been involved in improving editing performance. UBI10 was found to be more effective in the germline of *Arabidopsis* [31].

The key benefit of this method is that it does not require a plant-tissue culture unit. Moreover, it is cost-effective and easy. This method is extensively and commonly used for editing the *Arabidopsis* genome worldwide. The downside of this method is that it is limited to a small range of plants, such as *Linum usitatissimum*, *Arabidopsis*, and *Solanum lycopersicum*, with less productivity due to the limited formation of seeds and flowers.

#### 3.2.2. Agrobacterium-Mediated Delivery 

*Agrobacterium*-mediated delivery methods are the most frequently used delivery methods for a wide variety of plants. The binary vector containing Cas9 and the gRNA expression cassette is transformed into the *Agrobacterium* strain. In addition, the *Agrobacterium*-mediated genetic transformation of CRISPR constructs the required explants, such as floral organs, calluses, and leaves. *Agrobacterium*-mediated transformation is more effective and has a higher editing efficiency than the particle bombardment process.

Monocot crops with lower regeneration and transformation abilities are widely used in *Agrobacterium*-based transformations. In one study, 100% editing efficiency was recorded in the Cavendish banana cultivar “Williams” using the *Agrobacterium*-mediated gene transformation [32]. Among all the delivery methods, *Agrobacterium*-mediated delivery is the most successful and effective approach for woody plants. Woody plants such as *Citrus sinensis* and *Populus* have been effectively modified via *Agrobacterium*-mediated genome editing [33].

The major advantage of this approach is the high performance of editing compared to other known approaches. Another advantage is its broad applicability and the fact that it is readily accessible and less expensive than some other approaches. Stable transgene integration through single-copy integration can be accomplished with this approach. The main downside of this approach is that it involves a binary vector that integrates a foreign gene into the genome of the plant.

## 4. Nanoparticle-Based Carriers in CRISPR/Cas

Nanoparticle-based carriers have been used for delivery of CRISPR constructs into targeted cells (Figure 2). Several reports of nanoparticle-based delivery of CRISPR/Cas have been elaborated in Table 1. The paragraphs below discuss the properties of all these nanoparticle-based carriers in detail.

### 4.1. Exosome-like Nanoparticle-Mediated Delivery of CRISPR/Cas Cargos

Exosomes are nanoscale membrane vesicles with a diameter range from 30–100 nm that are secreted by almost every kind of cells and stably exist in virtually all kinds of bodily fluids [34]. They can transmit a variety of signaling molecules, including nucleic acids (mainly mRNA and microRNA), functional proteins, and lipids [35,36,37]. Owing to the small size of exosomes, they can escape from the rapid phagocytosis caused by mononuclear phagocytes, steadily carry and deliver drugs in circulation, and pass-through vascular endothelium to target cells [38]. Exosomes have been proven to increase the stability of their content and, therefore, they can play a role in enhancing the bioavailability of bioactive compounds [39]. Exosome vesicles are composed of proteins, lipids, and nuclear components that accumulate randomly from the cells in which they originate [40]. Moreover, it is likely that these vesicles already function as nanocarriers within the plant and animal tissues. Despite the brief period of time since the discovery of these organic nanocarriers, there have been reports that plants transfer sRNA to pathogens and pests to inhibit their virulence. Recently, researchers have found new delivery tools for the trafficking of these RNA molecules from plants to the pathogen. These tools are exosome-like extracellular vesicles, which move across the kingdoms from organism to organism and cause silencing. In *Arabidopsis* and *Botrytis cineria* interactions, the host cell delivers sRNAs into fungal cells using extracellular vesicles [41]. These interesting findings may facilitate the use of these extracellular vesicles for the effective delivery of all kinds of sRNAs to silence pathogen virulence genes and provide resistance [42].

For further clarification, the endosomal pathway is an evolutionarily conserved membrane-trafficking mechanism important for recycling and degradation of plasma membrane proteins. Starting with endocytosis, early endosomes are formed by inward budding of the plasma membrane and mature into late endosomes. As maturation progress, intraluminal vesicles bud and form inward multivesicular endosomes (MVEs) [43]. Maturing endosomes have different fates: they fuse with the vacuole and lead to cargo degradation or fuse with the plasma membrane, releasing its luminal contents. The intraluminal vesicles of MVEs are released as exosomes. Important regulators of intracellular membrane trafficking include small GTPases, specific subsets of which mark membrane compartment identity [44]. Extracellular vesicle (EV)- and MVE-like structures have also been observed in plants using microscopy techniques at infection sites of fungal pathogens [44,45,46]. Ultra-structural examination of non-host interactions between the barley powdery mildew fungus *Blumeria graminis* f. sp. *hordei* and *Arabidopsis* revealed plant MVEs and syntaxin PEN1-positive exosomes accumulating around the fungal infection structures [47,48,49]. 

### 4.2. Delivery of CRISPR/Cas Reagents through Liposome Nanoparticles

The use of liposomes is one of the methods based on the different types of nanoparticles. Liposomes are sphere-shaped vesicles that carry a compound of interest and consist of one or more phospholipid bilayers. The plant cuticle has amphiphilic properties and small size ranging between 80 and 300 nm [44,50]. Liposomes can offer several advantages as vectors for gene delivery into plant cells, including enhanced delivery of encapsulated DNA by membrane fusion, protection of nucleic acids from nuclease activity, targeting to specific cells, and delivery into a variety of cell types besides protoplasts via entry through plasmodesmata [51,52]. In liposome-based gene therapy there is no potential toxicity for humans and plants [53,54]. 

Today, liposomes are used as reagents and tools in various scientific disciplines. Since liposomes have many attractive features, they have made their way onto the market. In the cosmetics and pharmaceutical industries, numerous molecules act as carriers; moreover, in the food and farming industries, liposomes are used in encapsulation to grow delivery systems that can entrap unstable compounds [55]. Liposomes have on many occasions served as model systems for cellular membranes to investigate protein functionality regarding osmotic and pH stability. 

In addition, liposomes have been investigated by researchers regarding plant aging, drying tolerance, freeze tolerance, the presence of toxins that work against pesticides, and even many industrial food applications [56]. Moreover, monitoring of the organo-phosphorus pesticides dichlorvos and paraoxon at very low levels has been achieved with liposome-based nano-biosensors [57]. Liposomes have been used as transfection agents to deliver various materials to cells [58,59]. Liposomes consist of vesicles, bounded by a lipid bilayer, that can be loaded with cargo, i.e., nucleic acids and/or proteins [60,61,62,63]. They have also been used to deliver nitric oxide in plant cells [64].

Cationic liposomes are incorporated into membranes to deliver NO intracellularly. Being encompassed by liposomes, the activity of nitric oxide is stabilized and withheld until the point of application. Gaseous NO, when encapsulated in an echogenic liposome, remains active up to 37 °C in the serum, which is suitable for prolonged uses [64]. Many reports have verified this idea of developing an effective liposomic photo-stimulated NO donor in plants [65,66].

**Table 1 ijms-22-07456-t001:** Successful reports of nanoparticle-based delivery of CRISPR/Cas.

Performed Editing	Targeted Gene	Delivery Method	Reference
Gene knockout	*BAFFR*	Polyethyleneimine–cyclodextrin	[67]
Gene knockout	*PLK-1*	Liposome-templated hydrogel nanoparticles	[68]
Gene knockout	*PD-11*	Human serum albumin (HSA) nanoparticles	[69]
Homology-directed repair	*CXCR4*	Gold nanoparticles	[70]
Gene knockout	*EGFP*	DNA nanoclew	[71]
Gene knockout	*VEGFA*	Aptamer-functionalized lipopolymer	[68]
Gene knockout	*PD-11 gene*	Human serum albumin (HSA) nanoparticles	[69]
Gene knockout	*Ptch1, Trp53, Pten*	PEI	[72]
Gene knockout	*AAVS1*	Gold nanoparticles	[73]
Gene knockout	*mCCR7*	GNOME	[74]
Gene knockout	*CD38*	Nano-blade chip	[75]
Gene knockout	*GFP*	Bioreducible lipidoid nanoparticles (LNPs)	[74]
Gene knockout	*YFP*	Gold nanoparticles	[76]
Gene knockout	*H11*	Core-shell NPs	[77]
Gene knockout	*EGFP*	Bioreducible lipids	[78]

Moreover, Arguel et al. [79] used the transfection reagent lipofectamine in nematodes to improve siRNA uptake from the environment. However, this protocol failed in *A. rhodensis,* prompting its combination with microinjection. In addition, Adams et al. [80] stated that by combining liposome-based technology with microinjection into the gonad, functional genomic techniques become feasible in non-model nematode species. With this method, RNAi and CRISPR/Cas9 mutagenesis become extremely efficient in *A. rhodensis* and *A. freiburgensis*. 

### 4.3. Exosome/Liposome Hybrid Delivery of CRISPR/Cas Reagents

To improve the loading capability and delivery efficiency of exosomes, hybrid exosome-like and membrane-camouflaged nanovesicles, like synthetic nanocarriers with natural exosomes, can be designed. In addition, targeted delivery of the CRISPR/Cas9 system to the receptor cells is essential for in vivo gene editing [81]. Recently, exosomes have been intensively studied as a promising targeted drug delivery carrier, although they are limited by their low efficiency in encapsulating large nucleic acids. Lin et al. [81] prepared a kind of hybrid of exosomes and liposomes through simple incubation. Different from the original exosomes, hybrid nanoparticles efficiently encapsulate large plasmids, including the CRISPR/Cas9 expression vectors, similarly to liposomes. 

The advantages of exosomes over other synthetic carriers concern their small size, low toxicity, natural targeting ability, their ability to encapsulate various endogenous bioactive molecules, and their ability to cross many physical barriers. Therefore, exosome-based nanocarriers may have more chances of success as vehicles of drugs, DNA, oligonucleotides, proteins, peptides, etc. With the issues and limitations being addressed, exosome-based nanocarriers can promote the advancement of an effective drug delivery system. The collection of exosome-like nanoparticles from high-efficiency plants can be a good alternative to other methods, since they are easier to extract and do not have the drawback of being toxic in animal cells. Natural and synthetic exosome-like nanoparticles, produced from serial extrusion of cells or by bottom-up synthesis, are currently some of the most promising, high-efficiency, biocompatible systems for drug delivery [82].

Nano-biotechnologists have harnessed the untrained power of genome editing to develop crops. Combinations of nucleic acid-templated organic or inorganic nanomaterials and molecular genetics can result in fairly inexpensive methods for the food industry, agriculture, industrial biotechnology, and other sectors associated with the bio-economy. Exosomes can be explored as a stable and natural nano-delivery system for increasing the bioavailability of these compounds. Moreover, it is likely that these vesicles function as nanocarriers within plant and animal tissues [39]. Researchers have attempted different approaches to encapsulate the CRISPR–Cas9 system into exosomes and found the proposed hybrid exosome produced via incubation with liposomes to be an effective new strategy for drug encapsulation and for delivering the CRISPR–Cas9 system in vivo [81]. However, more research is needed on the biogenesis of plant-derived exosome-like vesicles and how they respond to various environmental conditions in order to enhance their production and extraction yields for large-scale food applications. Furthermore, the role of exosomes as delivery vesicles within the plant and animal systems needs clarification. 

Interestingly, recently reported artificial multi-layer NPs exhibited considerable delivery efficiency with CRISPR/Cas9 systems, demonstrating the superiorities of multi-layer structures in delivering CRISPR cargos [19]. The multi-layer structures of these NPs are quite similar to viral vectors; for example, they have (1) a cationic core (organic or inorganic) to stabilize the CRISPR cargos; (2) an organic shell (mainly a lipid layer) to protect the inner complexes; and (3) multifunctional ligands anchored on shell to facilitate the bio-interactions with host cells. 

CRISPR cargos can be efficiently and stably encapsulated into the multi-layer NPs, and the material compositions can significantly influence the extra-/intracellular fate of these cargos and thereby the genome-editing efficiency. To transport CRISPR tools into the nucleus so that they can exert their biological functions, an ideal non-viral vector needs to load the cargos inside of cells efficiently, accurately deliver the cargos to the specific site, and further mediate the efficient cellular endocytosis of cargos. 

The carrier also needs to help the cargos break through the intracellular barriers, release the cargos in a timely manner, and facilitate their entrance to the nucleus. Of course, the stability and bio-activity of the cargos must be ensured throughout the processes. This might be achieved by selecting the non-viral materials and designing the structure of the multi-layer NPs appropriately. It also highlights the possibilities of converting the current potential building blocks into multi-layer NPs while maintaining their delivery capabilities. Multi-layer NP-based CRISPR/Cas9 delivery is efficient and safe. With regard to efficiency, the targeted delivery into intended tissues and specific cells must be achieved. The biggest challenge is to specifically and efficiently deliver cargos into target sites using multi-layer NPs, which is crucial for successful clinical transformation. In this regard, the recently reported triple-targeting strategy may provide a feasible approach to enhance the efficiency of target delivery in vivo [83]. 

Thus, a more in-depth understanding of nano-bio interactions is still needed, for instance, of: (1) the interactions of nano-systems with and the causes of their being taken up by mononuclear macrophage systems, (2) the dynamic mechanism between non-viral vectors and targeting sites. Secondly, the CRISPR–Cas9 vector must accurately be delivered into the nucleus for gene editing. After the specific localization at target sites, cellular uptake, endosomal escape, cytoplasmic transport, and nuclear import are necessary steps for successful delivery, which is much more complex and difficult compared to traditional nucleic acid delivery. 

The positive surface charges of commonly used vectors may facilitate the cellular uptake of NPs, but the specificity needs to be cautiously considered. Incorporating targeting molecules, antibodies, or aptamers on the surfaces of NPs seem to be a feasible approach to improve specificity. However, modifying NPs with targeting moieties also increases the difficulties of incorporating additional components inside the vectors [84].

Efficient cargo release inside cells in a spatiotemporal manner is another important issue. Otherwise, gene-editing tools will not work within a specific time window, which can result in unexpected off-target effects. Effective strategies, such as controlling the activity of Cas9 protein with a small molecule, have been established. 

Researchers have used materials with disulphide linkages to build delivery systems and they achieved GSH-triggered release of gene-editing tools, which further enhanced the editing efficiency. Moreover, the photo-thermal conversion abilities of gold NPs were fully utilized to achieve intracellular thermos-triggered plasmid release under the stimulation of an external laser. These methods offer approaches for the spatiotemporal control of the cargo release and further improve the efficiency and accuracy of CRISPR/Cas9 systems through the optimization of the applied materials. Following the cargo release, the CRISPR/Cas9 system can be imported into the nuclei under the guidance of incorporated nucleus targeting ligands (such as TAT and NLS). 

In most cases, cationic materials are utilized to build multi-layer NPs for the delivery of CRISPR/Cas9 systems; however, safety is a major concern. Cationic materials possess some intrinsic drawbacks, including the destabilization of the cell membrane, which may induce serious cytotoxicity. In addition, cationic vectors, such as cationic liposomes, are mainly internalized into cells via endocytosis, followed by entrapping in endosomes or lysosomes; however, the acid environment easily degrades cargos and thereby reduces transfection efficiency. The mononuclear phagocytic system acts as a scavenger of the cationic systems during blood circulation, which significantly hinders the accuracy of cargo transportation. 

Recently, the successful employment of non-cationic materials for in vivo CRISPR/Cas9-based gene therapy of triple negative breast cancer (TNBC) demonstrated an alternative strategy for cationic materials. Moreover, the biodegradability of used materials should be valued. TNBCs and their metabolic products have been well-studied, showing clear metabolic pathways that ensure biocompatibility and biosafety. This approach can greatly shorten the development cycle and accelerate the clinical application. On the other hand, the CRISPR/Cas9 system itself has the risk of an off-target effect, which means unexpected editing in the genome of targeted cells. To alleviate this, rationally designing the sgRNA and deliberately choosing the targeting site seem to be effective approaches. Moreover, Cas9 mRNA and protein formats are superior to those of Cas9 plasmids from the perspective of the reduction of off-target effects, which may induce higher efficiency for double-stranded break (DSB) formation with the help of non-viral carriers. 

Moreover, by fully utilizing the properties of these multi-layer NPs, even remotely controllable editing systems can be developed. In short, it is feasible to learn from the viral vectors to improve the efficiency of delivery and editing, and multi-layer NPs can be regarded as a series of biomimetic carriers of viral vectors. Using these NPs to deliver the CRIPSR cargos can not only overcome the safety concerns caused by viral vectors but also improve the variability of non-viral vectors for the delivery of the CRISPR/Cas9 system [85,86]. 

Nanovehicle technology has the potential to address the challenges of feeding the exponentially increasing world population under extreme environmental conditions in terms of the biotic and abiotic stresses, as well as through plant modifications towards pest- and disease-resistance traits [87]. This approach results in nanoparticles that possess defined nanostructures as well as multiple functions that may enable them to overcome the critical extracellular and intracellular barriers to successful gene delivery [88]. The development of nanovehicles as molecular transporters could become a key driver for genetic transformation of plants by resolving the delivery challenges and enhancing the efficacy of plant genetic engineering.

### 4.4. Nanoclew-Mediated Delivery of CRISPR Reagents

Recently, CRISPR/Cas9 was identified as a fascinating GE tool for targeted genome manipulations and for the expression of desired genes in numerous organisms [89,90]. The CRISPR/Cas9 system has emerged as the most powerful tool for GE in several organisms, including model and non-model plant species [91]. The latest ground-breaking technology involving CRISPR/Cas9 is basically an adaptive immune system in type II prokaryotes that protects them against invading organisms during phage infection by spacer acquisition, biogenesis, and target degradation [92].

The toolbox of CRISPR/Cas9 was adapted from bacteria as well as *Archaea* and includes engineered nucleases [93,94]. There are two main components of the CRISPR/Cas9 system: a single-guide RNA (sgRNA) that identifies a specific DNA sequence and the Cas9 protein that produces DSBs at a targeted site [92]. Therefore, by changing the design of the sgRNA, numerous desired sites can be targeted, which makes CRISPR/Cas9 simpler to handle than TALENs and ZFNs [95].

A DNA “nanoclew” is a unique technology for CRISPR/Cas9 component delivery. Developed by Sun et al. (2014), a DNA nanoclew is a sphere-like structure of DNA that has been compared to a ball of yarn. The nanoclew is synthesized by rolling circle amplification with palindromic sequences that aid in the self-assembly of the structure. The sphere can then be loaded with a payload that can be specifically triggered for release by certain biological conditions. As DNA nanoclews are a new delivery technology, they have currently only been utilized in an in vitro setting. In 2015, a group re-purposed the nanoclew for CRISPR/Cas9 delivery by designing the palindromic sequences to be partially complementary to the sgRNA within the Cas9:sgRNA ribonucleoprotein complex (Sun et al., 2015) [77].

By coating the nanoclew with polymer polyethylenimine (PEI) to induce endosomal escape, the group demonstrated roughly 36% efficiency in the delivery of CRISPR/Cas9:RNP with the nanoclew (compared with 5% with bare Cas9:sgRNA and PEI). This allowed the nanoclew to attain efficiencies comparable to other high-efficiency CRISPR/Cas9 delivery systems but still contain no viral components (or, indeed, any exogenous material besides repeating DNA and PEI) [96].

### 4.5. Cationic Lipid Nanoparticles

Cationic lipid nanoparticles are powerful colloidal carriers. Their advantage over other colloidal carriers made up of non-organic material or polymers is the use of biocompatible lipids. This makes these systems similar to real-life systems [97]. The notable benefit of cationic lipid nanoparticles over polymeric NPs is that they can be generated without the use of organic solvents using a high-pressure homogenization (HPH) process [98]. Cationic lipid nanoparticles with at least one cationic lipid have been described as non-viral vectors for delivery of genes [99]. 

The uses of cationic lipid nanoparticles in these kinds of applications are already well-known [100,101]. It has been observed that they can effectively bind nucleic acids, shield them from breakdown of DNAase I, and transfer them to living cells [102,103,104]. Initial evidence of the in vivo efficacy of cationic lipid nanoparticles has also been published [105]. Scholars have researched the materials and production processes involved and conducted in vitro experiments [99].

While satisfactory proof has been obtained of these nanoparticles’ effectiveness in the transmission of DNA/RNA to living cells [99,105], there are still several features that need to be clarified. The predictable stability and polymorphic transitions of cationic lipid nanoparticles have been associated with their solid-state [106].

## 5. Comparison of Exosomes/Liposomes with Other Carriers

There are many key differences that distinguish exosomes/liposomes from other carriers. These differences relate to the price, time consumption, and efficiency of the process. The key benefit of using NPs in CRISPR/Cas9 delivery is the flexibility with which they can be designed, and the properties of nanoparticles can be tailored to conform to CRISPR/Cas9. Moreover, liposomal carriers can carry complex payloads due to the incorporation of targeting moieties on the surface of the liposome. The synthesis of an aptamer–liposome–CRISPR/Cas9 chimaera approach, which led to improved and accurate delivery of CRISPR/Cas9–gRNA in vitro with reduced immunogenicity relative to other cationic liposomal formulations, was facilitated by the modification of liposomal surfaces with aptamers [107]. 

One of the most widely used methods for CRISPR/Cas9 delivery is viral delivery. Despite their numerous benefits, such as precision in genome editing, viral delivery systems have some significant shortcomings, e.g., the possibility of insertional mutagenesis, sequence insertion limitations, a long and difficult large-scale procedure, and immune system activation, which can impede their therapeutic application [108,109]. Synthetic lipid NPs are among the better carriers for CRISPR/Cas9 systems due to their reduced toxicity, the safety of the CRISPR/Cas9 vector or sgRNA from nucleases, and a significant decrease in immune response activation [110]. The cationic liposome benefits from an interacting negatively charged cell membrane, which makes it possible to encapsulate nucleic acids [111]. Comparison of exosomes/liposomes with other carriers of CRISPR/Cas has been provided below in Table 2.

## 6. Emerging Delivery Tools for CRISPR Cargos in Plants

There are various emerging tools for delivery of CRISPR cargos in plants. There are three basic CRISPR reagents that have been frequently used for genome editing (Figure 3). Nanotechnology is advancing with each passing day; moreover, the other related technologies are achieving perfection in the delivery of CRISPR cargos. One such example is pollen magnetofection-mediated genome editing. Magnetofection is a genetic transformation technique that uses magnetic force to absorb a vector connected with magnetic nanoparticles (MNPs). This approach uses positively charged Fe_3_O_4_-coated polyethylenemine MNPs and negatively charged vectors to form MNP-DNA complexes. Furthermore, the pollens are combined with the complexes and the magnetic field is implemented and applied for pollination. This technology has been applied effectively for cotton.

Currently, two approaches are commonly used for distributing CRISPR/Cas9 components: the first uses CRISPR/Cas9 vectors, and the second one uses a vector-less CRISPR/Cas9 method. Magnetofection is based on vector-free editing which is useful for the generation of non-transgenic crops.

To date, there has been no research on the editing of genome-mediated pollen magnetofection. However, the benefit of this approach is that it makes it possible to directly transfer the CRISPR/Cas9 ribonucleoproteins to pollen. This technique can reduce the time needed for tissue culture and transgenic organism selection.

## 7. Challenges in Exosome/Liposome-Based CRISR Delivery

PEGylation is one of the challenges of liposomal CRISPR delivery. In principle, cleavable PEG is a perfect option for the PEG problem and the ABC phenomena caused by repeated PEGylated formulations. Though cleavable PEGylated liposomes outperform uncleavable PEGylated liposomes in both in vitro and in vivo experiments, the cleavage efficiency is unsatisfactory and current cleavable PEGylated liposomes are still less effective than non-PEGylated ones under in vitro conditions. Therefore, research on super-cleavable PEG is clearly needed [142]. In addition to other limitations, the precise cellular uptake process of cationic formulations, including lipopolyplexes and lipoplexes, is still undetermined. It could be simply non-specific ionic interaction-mediated endocytosis, or it could be influenced by other factors [142]. Furthermore, it is also unknown how much of the uptake through CRISPR/Cas9 successfully reaches the cytosol from endosomes, and the locations of unsuccessful CRISPR/Cas9 molecules in the cell are also difficult to determine. Advanced cell trafficking techniques are needed to understand the comprehensive mechanism, which could help with further enhancement of cationic formulation.

Focus is still required on the improvement of lipid nanoparticles to transport CRISPR/Cas9 components due to a lack of viral components. Enhanced packaging of CRISPR/Cas9 materials can improve the chances of any subset of packaged molecules achieving proper delivery. Better decorations on the surfaces of liposomes aid the targeting of particles to relevant cells or tissues, prevent detection by the immune system, facilitate endosomal escape, and improve packing of CRISPR/Cas9 components for the delivery of particles to tissues or cells [96]. The effectiveness of electroporation and lipofectamine formulations in eleven cell lines was studied recently. In this research, electroporation transfection demonstrated better efficiency than lipofectamine transfection [143]. Moreover, in the case of liposomal CRISPR delivery, all internal and external barriers must be taken into account. Since they pass through the cell’s membrane, the NPs are typically coated in an endosome. The cell wall quickly guides encased substances into the lysosomal pathway, allowing all lysosome components to be degraded. As a result, the cargo becomes capable of avoiding the endosome.

In addition, if the CRISPR complex can escape the endosome, it can then enter the nucleus, which may be a source of failure. As a result, high efficacy in the delivery of CRISPR/Cas9 components via lipid nanoparticles is uncommon. Although scientists have been able to achieve 70 percent in vitro alteration efficiency in cells, this was only possible after an extensive search for the best lipids for liposomes [96].

## 8. Future Prospects

Multiple plant pathosystems have been shown to involve the exchange of small RNAs between cellular organisms during infections/infestations of plant hosts. Exploiting this small RNA transfer may prove to be a useful strategy to engineer crops with improved disease resistance [144]. Despite all the NPs, translocation to the nucleus is a significant problem because, during the passage, RNAs may come across the lysosomal pathway, which causes the degradation of the lysosome material and leads to failure; therefore, this method of delivery has significantly lower efficiency. However, it can be used after some modifications [145,146]. The CRISPR technique has been successfully employed in agriculture to combat biotic and abiotic stresses and improve plant nutrition and yield [147]. Thus, it is essential to improve various aspects of nanovesicles [148]. For such nanomaterials, techniques and gene delivery approaches continue to be developed, and the key challenge is to balance transfection efficiency, targeting specificity, particle size, biodegradability, and cytotoxicity, as well as their short- and long-term fates in the environment [149].

## 9. Conclusions

Genetic engineering of plants may be able to address the crucial challenges of cultivating sufficient food and producing natural product therapeutics for an increasing global population under changing climatic conditions. Despite advances in genetic engineering across many biological species, the transport of genetic material into plant cells and nuclei remains the primary limitation for broad-scale implementation and high-throughput testing of genetic engineering tools, particularly for mature plants with walled cells. Additionally, regulatory oversight for genetically modified foods (GMOs) motivates the development of plant engineering technologies for which gene expression is transient and foreign DNA are not integrated into the host plant genome [1]. However, the most commonly used tool today for plant genetic transformations—*Agrobacterium*-mediated transformation technology—is unable to perform DNA- and transgene-free editing and yields random DNA integration. Similarly, DNA delivery methods that utilize a gene gun or other external forces, such as vertexing, can cause cell damage, which leads to increased rates of transgene integration, possibly due to the over-activation of the endogenous cellular DNA repair mechanisms commonly induced by stress and cell/DNA damage [150]. Biological nanocarriers, such as cell-penetrating peptide-displayed polyion complex vesicles (CPP-PICsomes), have also been used for DNA-free genome editing using Cas9 ribonucleoprotein [151].

The progress obtained with CRISPR/Cas in the last few years is breathtaking. Meanwhile, the progress in developing new gene delivery carriers has helped in the modification of organic nano-carriers, like exosomes and liposomes, which have unique properties, like size, charge, stability, and their low toxicity compared to the known viral vectors. Therefore, further research is required for the use of this technique in plants on large scales.

## Figures and Tables

**Figure 1 ijms-22-07456-f001:**
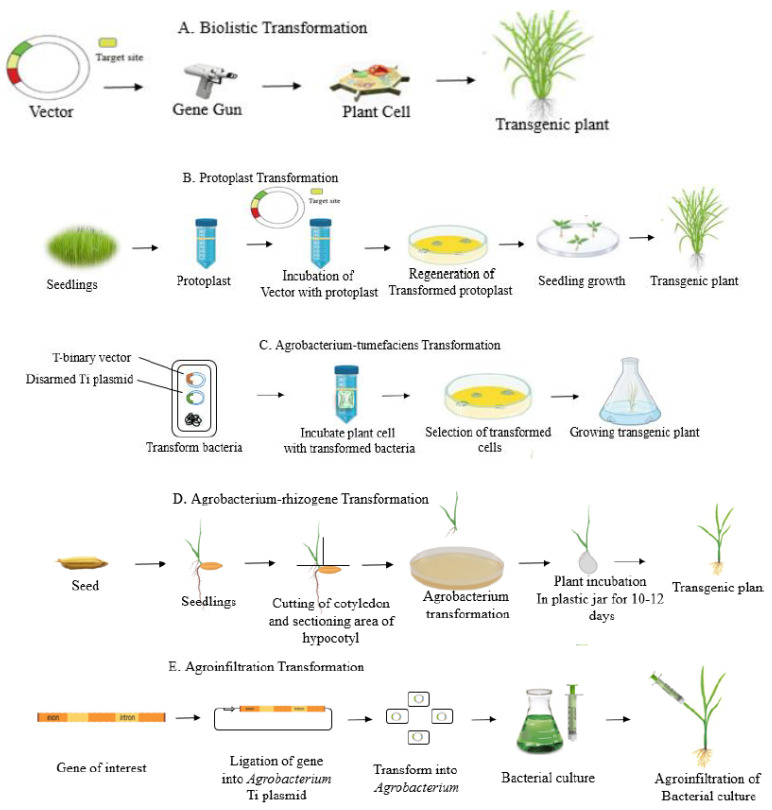
Delivery methods of CRISPR/Cas cargos into plants. (**A**) Biolistic transformation is based on the delivery of DNA into plant cells by high-velocity gold or tungsten particles with the help of a gene gun; (**B**) protoplast transformation involves the direct delivery of DNA into individual plant cells using polyethylene glycol or electroporation; (**C**) Agrobacterium-mediated transformations utilize the capability of the bacterial pathogen Agrobacterium tumefaciens to transfer foreign genes into a wide variety of host plants; (**D**) Agrobacterium rhizogene transformations transport single-stranded DNA (ssDNA; T-strands) and virulence proteins into plant cells through a type IV secretion system; (**E**) agroinfiltration transformations allow the suspension culture of agrobacterial cells to be infiltrated into the organs of an intact plant, providing a rapid and efficient way to transiently express foreign genes *in planta*.

**Figure 2 ijms-22-07456-f002:**
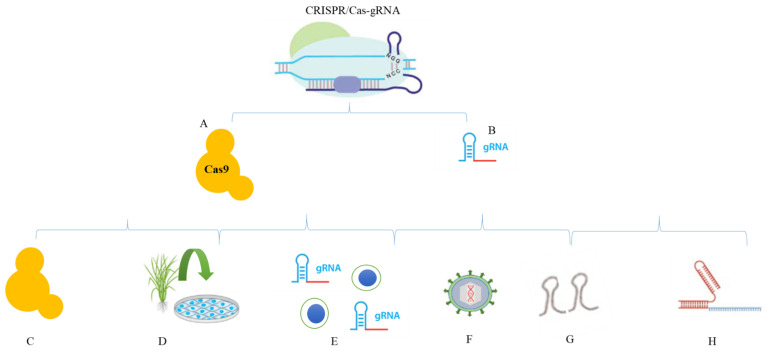
CRISPR/Cas9 cargos for delivery into cells. (**A**) Cas9 protein. (**B**) Trans-CRISPR RNA, sgRNA, and the Cas9–sgRNA complex can be delivered into cells using different methods. Some of the important methods are shown in the figure. (**C**) Cas9 protein delivery; (**D**) Cas9-expressing cell lines; (**E**) delivery of ribonucleoprotein (RNP); (**F**) delivery in the viral vectors/plasmids; (**G**) Oligos; (**H**) Delivery of the crRNA/tracrRNA complex as mRNA.

**Figure 3 ijms-22-07456-f003:**
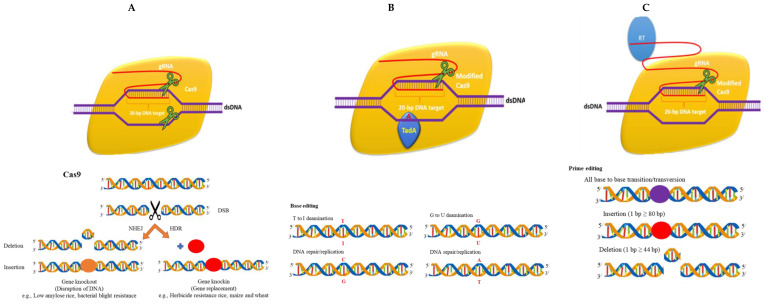
CRISPR/Cas reagents: (**A**) traditional Cas9 reagent; (**B**) CRISPR/Cas base editor; (**C**) prime editor.

**Table 2 ijms-22-07456-t002:** Comparison of exosomes/liposomes with other carriers of CRISPR/Cas.

CRISPR Carriers	Cargo	Composition	Capacity	Ease of Use	Advantages	Challenges	Reference
Liposome	mRNA; Cas9 and sgRNA; RNP	Natural/synthetic lipids and polymers	nM levels of sgRNA and Cas9	Low	- It is virus free - Manipulation is simple - Cost is low - Low immunogenicity	- Cargo endosomal degradation - Low efficiency in delivery - Low stability	[112,113,114,115]
Microinjection	DNA plasmid; mRNA; Cas9 and sgRNA; RNP	Needle	nM levels of sgRNA and Cas9	High	- Almost guaranteed delivery into targeted cell - High efficiency in in vitro experiments - Applicable for all CRISPR modes	- Time-consuming - Difficult to perform - Not suited for in vivo experiments - Cell damage - Costly equipment - Requires high expertise	[116,117,118,119,120,121,122]
Electroporation	DNA plasmid; mRNA; Cas9 and sgRNA; RNP	Electric current	nM levels of sgRNA and Cas9	Very Low	- Well established - High efficiency - Applicable for all CRISPR modes	- Generally, in vitro only - Some cells are not amenable - Damage to cells - Cytotoxicity	[123,124,125,126,127,128]
Adenovirus	DNA plasmid	dsDNA	8 kb nucleic acid	Moderate	- Highly efficient in delivery - Transient expression - Low rate of off-targets	- It can initiate an inflammatory response - Difficulty scaling production	[129,130,131,132,133]
DNA nanoclew	RNP	DNA spheroid	nM levels of sgRNA and Cas9	High	- Programmable - Controllable size and architecture - Non-toxic	- Requires template DNA modifications - Poor stability of the DNA carrier - Complicated assembly	[134,135,136]
Hydrodynamic method	DNA plasmid; RNP	High pressured injection	nM levels of sgRNA and Cas9	Low	- It is virus-free - Manipulation is simple - Cost is low	- Non-specific; it can be traumatic to tissues	[137,138,139,140,141]

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
