# Peer review of "Exosome/Liposome-like Nanoparticles: New Carriers for CRISPR Genome Editing in Plants"

_ijms, 2021, doi:10.3390/ijms22147456_

Round 1

Reviewer 1 Report

  1. The authors should work on the organization of their ideas, as the article feels like ideas are all over the place.
  2. Look for more citations, there are a number of long paragraphs with absolutely no citation.
  3. There are a number of incomplete/hanging sentences.
  4. Some of the paragraphs and sentences do do connect at all.
  5. Find a native English speaker to revise the English in the article.
  6. Choose and stick to one citation style, use either numbering or names but not both.

Author Response

Dear Reviewer,

Our response to your valued comments is attached

Reviewer 2 Report

This manuscript is a comprehensive review regarding CRISPR/Cas9 system in genetic engineering for plants. This review is well-written but has some grammatical errors and typos. I also suggest the authors add one following reference in Section 8 or 9, regarding recent technology on a direct delivery of Cas9/gRNA ribonucleoprotein in plants.

  1. Odahara et al. ACS Applied Nano Materials DOI: 10.1021/acsanm.1c00695

After minor revision regarding following points, I think the manuscript would be more attractive and suitable for publication as the review paper in IJMS.

As minor points:

  1. Page 3 Line 26, SpCas9 is misspelling?
  2. Page 3 Line 37, the abbreviations crRNA and tracrRNA should be explained at the first usage through the manuscript.
  3. Page 4, the bottom lines, I am not sure if the authors directly cite website addresses in the manuscript.
  4. Page 6, Line 24 and 34, reference numbering is weird (21x, 23x).
  5. Page 9, In Figure 2 caption, Mrna should read mRNA.
  6. Page 10, Line 33, a reference (Monstafa et al 2021) is not referred in a correct format.
  7. Page 12, in the second sentence: please revise spacing and the punctuation.

Author Response

(The authors gave the same response as above.)
